# Charge-neutral fermions and magnetic field-driven instability in insulating YbIr₃Si₇

Y. Sato [1,5], S. Suetsugu [1], T. Tominaga[1], Y. Kasahara[1], S. Kasahara [1,6], T. Kobayashi[1], S. Kitagawa [1], K. Ishida[1], R. Peters [1], T. Shibauchi [2], A. H. Nevidomskyy [3], L. Qian [4], E. Morosan [3,4] & Y. Matsuda [1✉]

Kondo lattice materials, where localized magnetic moments couple to itinerant electrons, provide a very rich backdrop for strong electron correlations. They are known to realize many exotic phenomena, with a dramatic example being recent observations of quantum oscillations and metallic thermal conduction in insulators, implying the emergence of enigmatic charge-neutral fermions. Here, we show that thermal conductivity and specific heat measurements in insulating YbIr₃Si₇ reveal emergent neutral excitations, whose properties are sensitively changed by a field-driven transition between two antiferromagnetic phases. In the low-field phase, a significant violation of the Wiedemann-Franz law demonstrates that YbIr₃Si₇ is a charge insulator but a thermal metal. In the high-field phase, thermal conductivity exhibits a sharp drop below 300 mK, indicating a transition from a thermal metal into an insulator/semimetal driven by the magnetic transition. These results suggest that spin degrees of freedom directly couple to the neutral fermions, whose emergent Fermi surface undergoes a field-driven instability at low temperatures.

[1] Department of Physics, Kyoto University, Kyoto 606-8502, Japan. [2] Department of Advanced Materials Science, University of Tokyo, Kashiwa, Chiba 277-8561, Japan. [3] Department of Physics and Astronomy, Rice University, Houston, TX 77005, USA. [4] Department of Chemistry, Rice University, Houston, TX 77005, USA. [5] Present address: RIKEN Center for Emergent Matter Science (CEMS), Wako 351-0198, Japan. [6] Present address: Research Institute for Interdisciplinary Science, Okayama University, Okayama 700-8530, Japan. ✉email: matsuda@scphys.kyoto-u.ac.jp

Strong electron interactions often lead to the emergence of many-body insulating ground states. Recently, surprising properties have aroused considerable interest in the research of the strongly correlated insulators, $SmB_6$ and $YbB_{12}$ with simple cubic crystal structures[1]. In these Kondo lattice compounds, the band gap opens up at low temperatures due to the hybridization of localized $f$ electrons with conduction electrons[2]. In particular, quantum oscillations (QOs)[3–7], specific heat[6,8,9], and thermal conductivity[6,8,10,11] experiments have posed a significant paradox, revealing gapless excitations in the bulk, in apparent contradiction with the charge gap seen in transport measurements. While the angular dependence of the QO frequencies suggests a three-dimensional (3D) bulk Fermi surface in $SmB_6$[4] and $YbB_{12}$[5], both materials remain robustly insulating to high magnetic fields (in $SmB_6$, a 2D Fermi surface has also been reported[3,12]). Various theoretical models of the QOs in these insulators have been proposed so far[13–22]. Another striking aspect is a nonzero low-temperature linear specific-heat coefficient $\gamma \sim 10\,mJ\,K^{-2}\,mol^{-1}$ for $SmB_6$[6] and $\sim 4\,mJ\,K^{-2}\,mol^{-1}$ for $YbB_{12}$[8,9] in zero field. As the specific heat is measured in the bulk insulating state, these results indicate the existence of gapless and charge-neutral excitations in the bulk consistent with an emergent Fermi surface of neutral fermions.

However, there are distinct differences in the gapless excitations in these correlated insulators. In $SmB_6$, the QOs are observed only in the magnetization (de Haas-van Alphen, dHvA, effect). The dHvA oscillations strongly deviate below 1 K from the Lifshitz-Kosevich theory, which is based on Fermi liquid theory[4]. In contrast, in $YbB_{12}$, the QOs are observed not only in the magnetization, but also in the resistivity (Shubnikov-de Haas, SdH, effect) and both dHvA and SdH oscillations obey the Lifshitz-Kosevich theory down to 50 mK[5]. Moreover, in $YbB_{12}$, a finite residual temperature-linear ($T$-linear) term in the thermal conductivity $K_0 \equiv \kappa/T(T \to 0)$ is observed, demonstrating the presence of gapless and itinerant neutral fermions[8]. On the other hand, $K_0$ in $SmB_6$ has been controversial. While $K_0$ of $SmB_6$ has been reported to be very small but finite[6], the absence of $K_0$ has been reported in[10,11].

A fascinating question is whether the QOs have any relationship to the neutral fermions. $YbB_{12}$ undergoes an insulator–metal transition at $\mu_0 H \sim 50\,T$, also confirmed by the recent SdH oscillation measurements[7]. By tracking the Fermi surface area, it has been revealed that the same quasiparticle band gives rise to the SdH oscillations in both insulating and metallic states. By using a two-fluid picture, it has been pointed out that neutral quasiparticles coexist with charged fermions[7]. In addition, it has been shown that $K_0$ depends on magnetic fields in $YbB_{12}$, suggesting that the neutral fermions can couple to magnetic fields[8]. These results suggest that the neutral fermions may be crucial for explaining the QOs in $YbB_{12}$ and other Kondo lattice insulators. Various theoretical models that invoke novel itinerant low-energy neutral excitations within the charge gap that can produce QO signals have been proposed, including Majorana Fermi liquids[17,18,22] and a spin liquid with spinon Fermi surface[15,16]. However, the nature of the neutral fermions is largely elusive and continues to be hotly debated. As the Kondo hybridization between magnetic moments and conduction electrons is the origin of the charge gap formation in these insulators, it is crucially important to clarify how the neutral fermions couple to the magnetic degrees of freedom. Thus, more systematic investigations on a new class of materials are highly desired to clarify the relationship between QOs, charge-neutral fermions, and magnetic properties.

Recently a new insulating Kondo lattice compound $YbIr_3Si_7$ has been discovered[23]. $YbIr_3Si_7$ has a trigonal $ScRh_3Si_7$-type crystal structure (Fig. 1a). The magnetization and neutron diffraction data show that Yb ions are very close to the trivalent state in the bulk[23]. In zero field, antiferromagnetic (AFM) order occurs below the Néel temperature $T_N = 4.0\,K$. Neutron diffraction measurements report[23] that, in the AFM state corresponding to the $\Gamma_1$ state, all the $Yb^{3+}$ moments are oriented along the crystallographic $c$ axis ([001]). Each $Yb^{3+}$ moment is aligned anti-parallel with its six nearest neighbors in the nearly cubic Yb sublattice and parallel with its co-planar next nearest neighbors. The ordered moment is $\sim 1.5\,\mu_B/Yb^{3+}$. We note that in $YbIr_3Si_7$, the number of free charge carriers has been suggested to be much fewer than the number of local moments[23]. It has therefore been proposed[23] that the system becomes insulating at low temperatures as all the free carriers are consumed in the formation of Kondo singlets. Thus, $YbIr_3Si_7$ has insulating bulk and long-range magnetic correlations, and is distinct from other simple Kondo insulators, such as $SmB_6$ and $YbB_{12}$. Interestingly, thickness analysis of the electric transport shows that $YbIr_3Si_7$ harbors conducting surface states whose origin is, however, not topological but rather has to do with the valence change to $Yb^{2+}$ near the sample surface[23].

In this paper, we investigate the low-energy excitations in the AFM insulating state of $YbIr_3Si_7$ by the low-temperature specific heat and thermal conductivity measurements. We find that both $\gamma$ and $K_0$ are finite at low fields, demonstrating the presence of mobile and gapless excitations of neutral fermions in the bulk insulating state, i.e., $YbIr_3Si_7$ is a charge insulator but a thermal metal. The AFM order of this compound can be widely tuned by the external magnetic fields. More precisely, the charge-neutral quasiparticle excitations are either gapless or gapped with an extremely small excitation energy gap, much smaller than the base temperature 90 mK of our thermal conductivity measurements. Most surprisingly, a spin-flop transition from AF-I to AF-II phase at $\mu_0 H \approx 2.5\,T$ gives rise to an opening of a tiny gap or a linearly vanishing density of states (DOS) of neutral fermions, indicating a transition from a thermal metal into an insulator/semimetal. These results suggest that spin degrees of freedom directly couple to the neutral fermions, whose emergent Fermi surface undergoes a transformation in applied field.

## Results
### Magnetic phases
*Resistivity.* Figure 1b depicts the $T$-dependence of the in-plane resistivity $\rho$ of $YbIr_3Si_7$ single crystals (#1 and #2) plotted on a log–log scale. At $T \sim 150\,K$, $\rho(T)$ changes its slope, which is attributed to the onset of Kondo correlations. Below $\sim 150\,K$, $\rho(T)$ increases rapidly with decreasing $T$. As shown in the inset of Fig. 1b, $\rho(T)$ increases exponentially as $\rho(T) \propto \exp(\Delta_c/k_B T)$ with the charge gap $\Delta_c \sim 5.9$ and $\sim 6.5\,meV$ for sample #1 and #2, respectively. At around $T_N$, $\rho(T)$ is suppressed and increases again with decreasing $T$ down to $\sim 0.3\,K$. Upon further reducing the temperature, $\rho(T)$ saturates down to the lowest temperature. Figure 1c depicts the low temperature resistivity in magnetic field applied parallel to the $c$ axis ($\mathbf{H}\|c$). The suppression of $\rho(T)$ at $T_N$ is reduced in magnetic field and is absent above $\mu_0 H = 3\,T$, consistent with the previously reported data[23].

It has been shown that the low-temperature saturation of $\rho(T)$ arises from the surface state[23]. In fact, the difference of the saturation values of $\rho$ between crystals #1 ($\sim 1\,\Omega cm$) and #2 ($\sim 2.1\,\Omega cm$) can be quantitatively explained by the area and thickness of the crystal planes used for the measurements. Similar phenomena have been reported in $SmB_6$ and $YbB_{12}$, in which the metallic conductivity takes place at the surfaces of the crystal, while electronic transport suggests the opening of a finite charge gap in the bulk at low temperatures. These metallic surface in $SmB_6$ and $YbB_{12}$ has been attributed to the topological insulating

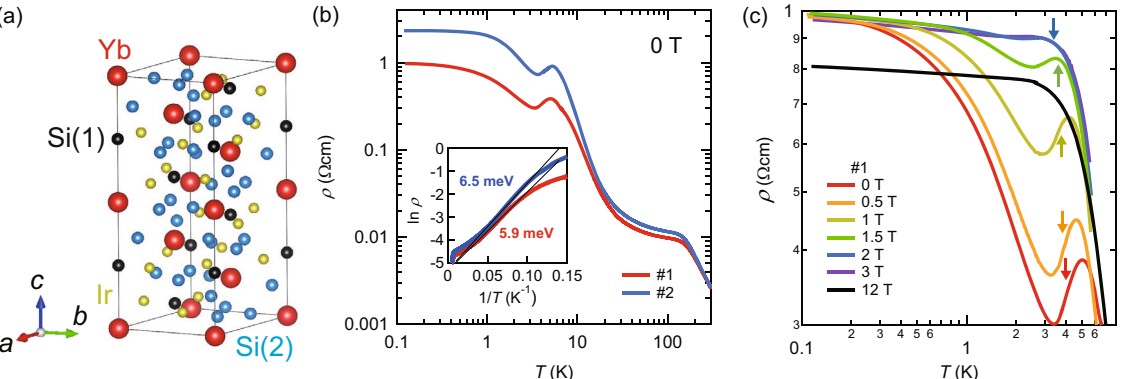

**Fig. 1 Crystal structure and transport properties of YbIr₃Si₇. a** Crystal structure of YbIr₃Si₇. There are two crystallographically inequivalent Si sites, Si(1) and Si(2). **b** Temperature dependence of the in-plane resistivity $\rho$ for #1 and #2 single crystals. The inset shows the Arrhenius plot of $\ln(\rho)$ vs. $1/T$. The solid lines represent the thermally activated behaviors with charge gaps of 5.9 and 6.5 meV for #1 and #2 single crystals, respectively. **c** Low temperature resistivity of #1 crystal in magnetic fields applied perpendicular to the *ab* plane. The arrows indicate the Néel temperature determined by the specific heat.

properties at low temperatures[24]. In fact, the metallic surface states have been resolved by angle-resolved photoemission spectroscopy (ARPES)[25,26]. In particular, spin-ARPES experiments in SmB₆ have revealed the spin-momentum locking of the surface quasiparticles as expected from topologically protected Dirac cones[25]. In YbIr₃Si₇, on the other hand, the recent photoemission spectroscopy measurements revealed that the surface conduction originates from a change of valence from Yb⁺³ in the bulk to Yb⁺² on the surface, without invoking topological arguments[23].

*Phase diagram.* Figure 2a displays the *T*-dependence of the specific heat divided by temperature, $C(T)/T$ of crystal #1 in zero and finite magnetic fields applied for **H**∥*c*. Specific heat shows a very sharp peak at $T_N = 4.0$ K in zero field. As indicated by arrows in Fig. 1c, the resistivity shows an anomaly at around $T_N$ determined by the specific heat. As depicted in Fig. 2a, $C/T$ increases on approaching $T_N$ from above. Similar phenomena have been observed in many antiferromagnets, such as CeRhIn₅[27]. This increase of the specific heat above $T_N$ is attributed the entropy release associated with the short-range AFM order or fluctuations. In YbIr₃Si₇, the magnetic field suppresses the peak height considerably and shifts $T_N$ to lower temperatures. The temperature dependence of $C/T$ changes dramatically at higher fields[23]. Above $\mu_0 H \approx 3$ T, $C(T)/T$ again exhibits a sharp peak, and the peak height increases rapidly, followed by a nearly saturated behavior above $\mu_0 H = 5$ T. In contrast to lower fields, $T_N$ is nearly independent of applied magnetic field. Figure 2b depicts $C/T$ plotted as a function of $T^2$ at low temperatures. An upturn of $C(T)/T$ at very low temperature ($T \lesssim 0.6$ K) is attributed to the nuclear Schottky anomaly of the Yb ions.

The specific-heat data clearly indicate the presence of two distinct AFM phases, i.e., low-field AF-I and high-field AF-II phases. To determine the phase boundary between these two phases below $T_N$, we measured the *H*-dependence of the magnetization *M* of crystal #3 taken from the same batch as crystal #1 for **H**∥*c*, as shown in Fig. 3a. At around $\mu_0 H \approx 2.5$ T, $M(H)$ curves show inflection points at low temperatures. To see this more clearly, we plot the field derivative of the magnetization d*M*/d*H* in Fig. 3b. At low temperatures, d*M*/d*H* shows a distinct peak as a function of *H*, which is attributed to the phase transition between the AF-I and AF-II phases. The peak field of d*M*/d*H* is independent of temperature. In addition, no discernible hysteresis is observed between up-sweep and down-sweep magnetization measurements. Therefore, the AF-I and AF-II phases are likely

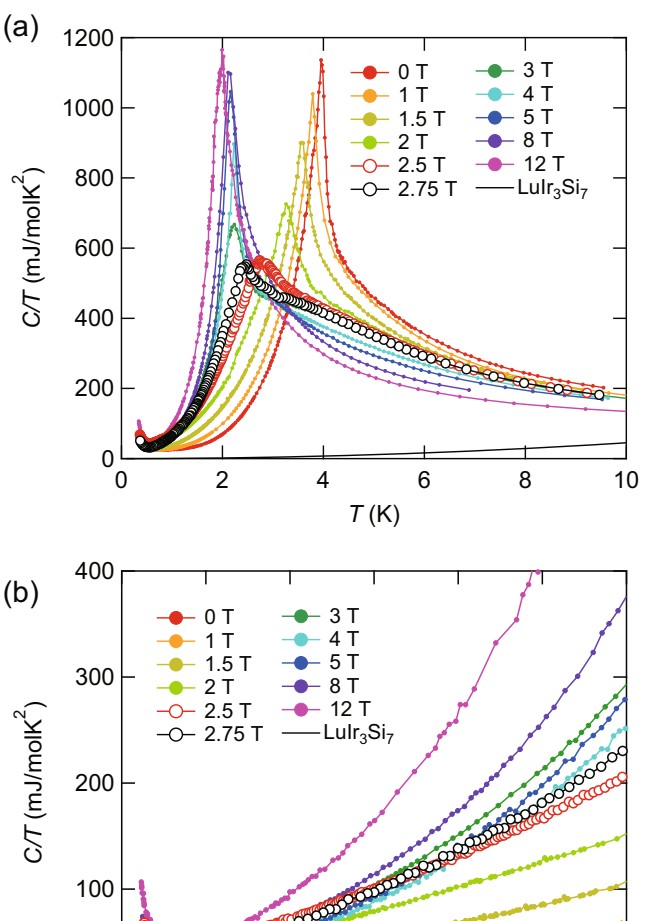

**Fig. 2 Specific heat of YbIr₃Si₇. a** Temperature dependence of the specific heat divided by temperature $C/T$ of YbIr₃Si₇ single crystal #1 in magnetic field perpendicular to the *ab* plane. Solid line represent $C/T$ of nonmagnetic and isostructural LuIr₃Si₇. **b** $C/T$ vs. $T^2$ at low temperatures in the magnetically ordered states.

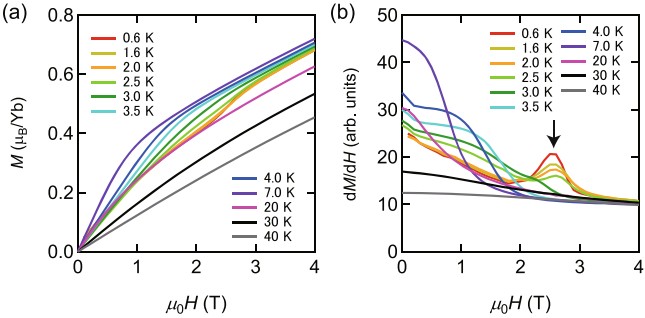

**Fig. 3 Phase boundary determined by magnetization measurements.**
**a** Field dependence of the magnetization $M$ of crystal #3 for **H** ∥$c$. **b** Field dependence of d$M$/d$H$. Peak indicated by the arrow corresponds to the boundary between AF-I and AF-II phases.

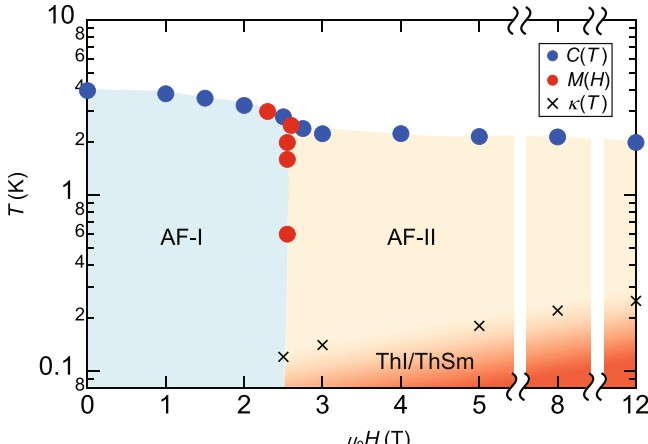

**Fig. 4 Field–temperature phase diagram of YbIr$_3$Si$_7$ for H ∥$c$.** The Néel temperatures (filled blue circles) are determined by the peak temperature of $C(T)/T$ and phase boundary (filled red circles) is determined by the peak of d$M$/d$H$. In the AF-I phase, the spins are oriented along the $c$ axis. The AF-II phase is in the spin-flop phase, where the spins are oriented in the $ab$ plane. The crosses represent the temperatures at which gap formation occurs, which is determined by the deviation of $\kappa/T$ from $T^2$-dependence shown by arrows in Fig. 8d–h. The red-colored regime represents thermal insulator or thermal semimetal (ThI/ThSm) regime.

separated by a weak first-order phase transition. Above 2.5 T, the magnetization increases gradually with $H$ without showing saturation. Moreover, as shown by the specific heat, a sharp phase transition is observed even above 2.5 T. In addition, the transition temperature slowly decreases with $H$. These results support that the AF-II is an AFM state, inconsistent with the ferromagnetic state.

Figure 4 displays the $H$–$T$ phase diagram for **H**∥$c$ axis determined by the specific heat and magnetization measurements. The Néel temperatures are determined by the peak temperature of $C(T)/T$. To obtain information on the nature of the phase transition, we performed nuclear magnetic resonance (NMR) measurements for **H**∥$c$ using another crystal (#4) taken from the same batch as crystal #1 and #3. Figure 5a, b depicts the magnetic-field swept $^{29}$Si-NMR spectrum in the AF-I and AF-II phases, respectively. There are two crystallographically inequivalent Si sites, Si(1) and Si(2), as illustrated in Fig. 1a. For comparison, the NMR spectrum at 4.2 K above $T_N$ are also shown by gray dotted lines. In the AF-I phase, the NMR spectrum splits into three peaks. The peaks in the higher and lower magnetic fields indicate that an internal magnetic field at the Si(2) site is parallel to the external magnetic field, which is shown in the inset

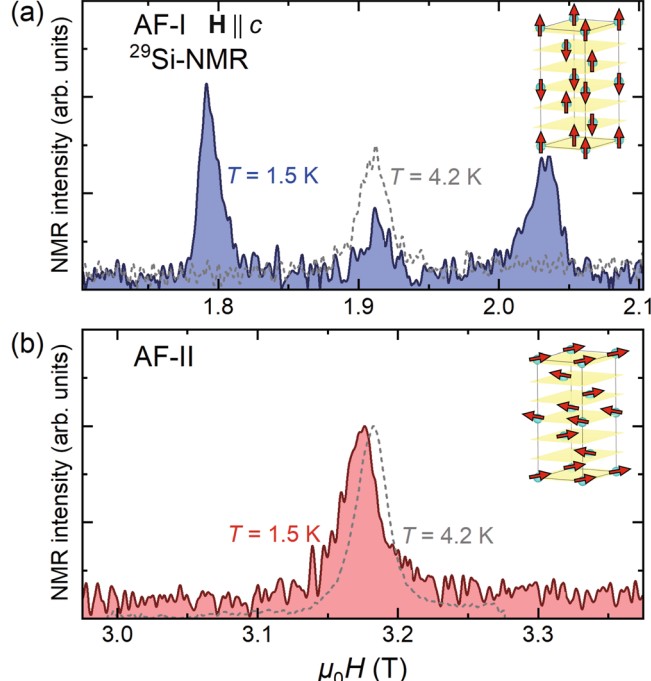

**Fig. 5 Magnetic structures revealed by NMR measurements.** Magnetic-field swept $^{29}$Si-nuclear magnetic resonance (NMR) spectrum for crystal #4 (**a**) in the AF-I phase and (**b**) in the AF-II phase. Gray dotted lines represent the NMR spectrum at 4.2 K (paramagnetic state). In the AF-I phase, the NMR spectrum split into three peaks, while in the AF-II phase, only one peak is observed. The NMR results indicate the spin-flop transition. The expected magnetic structure in each phase is shown in the insets. The yellow sheets represent planes of co-planar next nearest-neighbor Yb ions.

of Fig. 5a. This spin structure is consistent to that reported by neutron diffraction measurements. The middle peak arises from the Si(1) site at which an internal magnetic field from the Yb magnetic moment is canceled. On the other hand, in the AF-II phase, only one peak is observed. This peak slightly shifts to a lower field below $T_N$. This small shift suggests that dominant magnetic moments are oriented perpendicular to the external magnetic field, as shown in the inset of Fig. 5b, although the tilted angle from the $ab$ plane cannot be determined precisely in the present measurements. Thus, the NMR experiment reveals the spin-flop transition in which the magnetic moments oriented along the $c$ axis in the AF-I phase are rotated to the $ab$ plane in the AF-II phase.

**Gapless excitations in the insulating state**
*Specific heat.* The specific heat of nonmagnetic and isostructural LuIr$_3$Si$_7$ is plotted in Fig. 2a, b to estimate the phonon contribution. The phonon specific heat is negligibly small in the whole temperature range. As shown in Fig. 2b, $C(T)/T$ at low temperatures varies rapidly with $T$ at high fields. As the field is lowered, the $T$-dependence becomes weaker. Except for the very low $T$-regime, where $C(T)/T$ shows an upturn due to the nuclear Schottky anomaly of Yb ions, $C(T)/T$ increases with upward curvature with increasing $T$.

Figure 6a–h displays $C(T)/T$ vs. $T^2$ at low temperatures. Obviously, the extrapolation of $C(T)/T$ above 1 K to $T = 0$ yields finite intercepts for all fields, indicating the presence of a finite $\gamma$. The gapless magnon modes are expected to give rise to $C/T \propto T^\alpha$ with $\alpha = 1$ and 2 for 2D and 3D systems, respectively. In zero field, $C/T$ increases more steeply than $C/T \propto T^2$ line above 1 K.

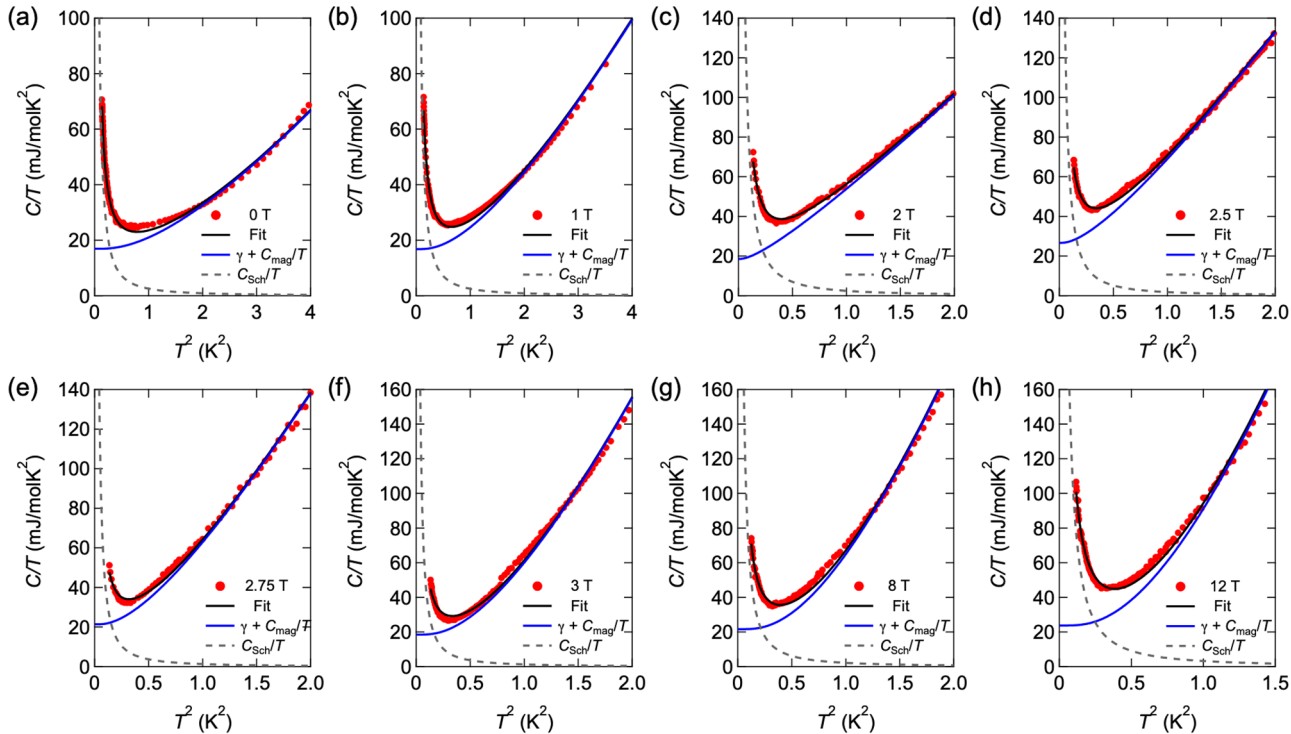

**Fig. 6 Fits to the specific-heat data. a–h** $C/T$ vs. $T^2$ for crystal #1 at several fields at low temperatures. The black solid, gray dashed and blue solid lines represent the total $C/T$, Schottky contribution, and $\gamma + \beta_M T^\alpha$ term, respectively, which are obtained by the fitting using Eq. (1).

Therefore, an additional source of specific heat (other than phonons, whose contribution has been subtracted) is required, which we associate with magnons. These magnon excitations are gapped even in zero field. This magnon gap is attributed to a large magnetic anisotropy of $Yb^{3+}$ ions due to the strong $LS$ coupling. As shown in Fig. 6a–h, the specific heat can be fitted by

$$\frac{C(T)}{T} = \gamma + \frac{C_{\text{mag}}(T)}{T} + \frac{C_{\text{Sch}}(T)}{T}, \qquad (1)$$

for all fields. Here, $C_{\text{mag}}(T) = \beta_M T^3 \exp(-\Delta_M/k_B T)$ is the magnon contributions with an excitation gap $\Delta_M$ and a coefficient $\beta_M$, and $C_{\text{Sch}}(T) = \frac{\Delta^2}{k_B T^2} \frac{e^{\Delta/k_B T}}{(1+e^{\Delta/k_B T})^2}$ is the two-level nuclear Schottky term, where $\Delta$ is the corresponding energy splitting. The low-temperature Schottky contribution is well fitted by $\Delta/k_B \approx 0.1$ K, which is field independent. Field dependence of $\Delta_M$ is shown in Supplementary Fig. 1. Near the boundary between the AF-I and AF-II phases, $\Delta_M$ is strongly suppressed. In the low-field regime in the AF-I phase and in high-field regime in the AF-II phase, $\Delta_M$ is nearly independent of $H$. In the ordered phase, $\Delta_M$ is determined by the competition between the Zeeman field and the molecular field due to the magnetic moment around the magnetic ions. When magnetic order is stabilized, the molecular field dominates and $\Delta_M$ is not seriously influenced by the Zeeman field. Because the magnetic order is suppressed and magnetic fluctuations are enhanced near the phase boundary, $\Delta_M$ is suppressed, consistent with the observed behavior of $\Delta_M$.

In Fig. 7a, the $H$-dependence of $\gamma$ obtained by the fitting of Eq. (1) is shown. In the whole-field regime, $\gamma$ is finite. In the low-field regime, $\gamma$ is nearly constant. Remarkably, $\gamma$ is enhanced above $\sim 2$ T, and peaks in the vicinity of the phase boundary. Upon entering the AF-II phase, $\gamma$ is first suppressed but then increases gradually with $H$. To confirm that this $H$-dependence of $\gamma$ is not due to a fitting ambiguity, we also plot $C(T)/T$ at 0.7 K, where the Schottky contribution is negligible. The similar

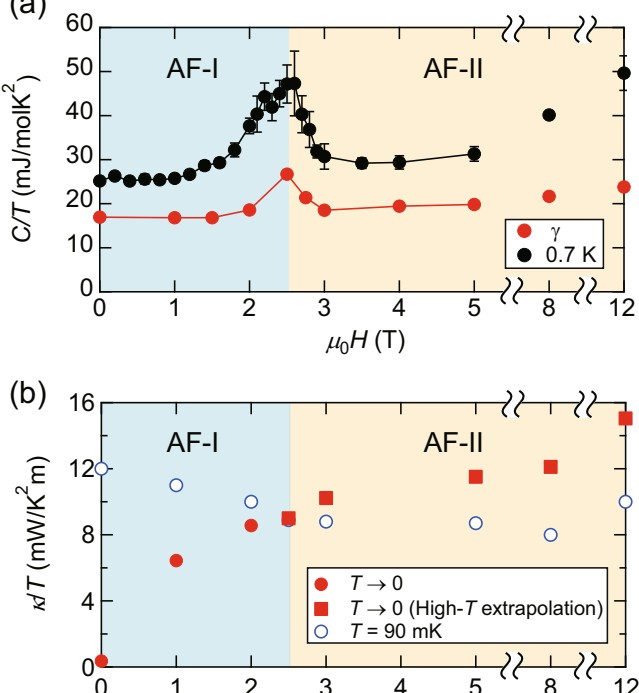

**Fig. 7 Field dependence of the fermionic excitations. a** Field dependence of $\gamma$, which is obtained by the fitting using Eq. (1) (see Fig. 6a–h), and $C/T$ at 0.7 K. The error bars represent standard deviation. **b** Field dependence of the residual thermal conductivity $K_0$ (filled red circles) in the AF-I phase and $\kappa/T$ obtained by the extrapolation from high-temperature regimes to $T = 0$ in the AF-II phase (filled red squares). Open blue circles represent $\kappa/T$ at 90 mK.

*H*-dependence of $C(0.7\,\mathrm{K})/T$ indicates that the enhancement of $\gamma$ near the phase boundary is an intrinsic property. As the system is insulating, finite $\gamma$ indicates the presence of a finite DOS of charge-neutral excitations. More precisely, the charge-neutral quasiparticle excitations in the AF-I and AF-II phases are either gapless or gapped with an extremely small excitation energy gap, much smaller than 0.7 K. To check the reproducibility of the data, we measured the specific heat of crystal #2 grown in the different batch. As shown in Supplementary Fig. 2, $C/T$ of #2 well coincides with that of #1, suggesting that the finite $\gamma$ is a universal property of this system. We shall discuss the field dependence of $\gamma$ in more detail below.

The strong suppression of the peak height of $C(T)/T$ and the reduction of $T_N$, and the magnon gap approaching the phase boundary between the AF-I and AF-II phases suggest a possible influence from a putative field-induced AFM quantum critical point (QCP). In fact, the magnitude of the magnetic moment is expected to be strongly reduced with approaching an AFM QCP, which leads to the suppression of the peak height of the specific heat, as reported in CeRhIn$_5$[27], and of magnon gap. In YbIr$_3$Si$_7$, however, the putative AFM QCP is avoided by a transition into the AF-II phase. Nevertheless, the quantum critical fluctuations emanating from an avoided QCP in the AF-II phase would lead to the reduction of the magnetic moment. These results lead us to consider that the enhancement of $\gamma$ in the AF-I phase near the phase boundary is caused by the AFM quantum critical fluctuations. The striking enhancement of $\gamma$ near the AFM QCP has been reported in several classes of strongly correlated electron systems, including heavy fermions[27] and iron pnictides[28]. The present results suggest that fluctuations emanating from an avoided AFM QCP largely modify the DOS of the neutral fermions.

*Thermal conductivity.* The specific heat involves both localized and itinerant excitations. Therefore, a finite $\gamma$ does not always indicate the presence of mobile gapless excitations. In fact, amorphous solids and spin glasses exhibit a finite $\gamma$, although the excitations in these systems are localized. Moreover, the Schottky anomaly in the specific heat often prevents the analysis of $C$ at very low temperatures. In contrast, the thermal conductivity is determined exclusively by itinerant excitations. In addition, it is free from the Schottky anomaly, enabling us to extend the measurements down to lower temperatures. In particular, a finite intercept $K_0$ provides the most direct and compelling evidence for the presence of the itinerant and gapless fermionic excitations, analogous to the excitations near the Fermi surface in pure metals.

Figure 8a–h shows $\kappa/T$ of crystal #1 plotted as a function of $T$ (main panels) and $T^2$ (insets) in zero and finite magnetic fields for $\mathbf{H}\|c$ at very low temperatures. In the AF-I phase, the $T$-dependence of $\kappa/T$ shows a convex downward curvature for $\kappa/T$ vs. $T$ plot, but a convex upward curvature for $\kappa/T$ vs. $T^2$ plot. As shown in Fig. 8d–h, the behavior of the thermal conductivity in the AF-II phase at $\mu_0 H \geq 2.5$ T is fundamentally different from that in the AF-I phase; the temperature dependence of $\kappa/T$ shows a concave downward curvature below ~0.3 K. As shown by dashed lines in the insets of Fig. 8d–h, $\kappa/T$ increases nearly proportional to $T^2$ in the high-temperature regime. At very low temperatures, $\kappa/T$ deviates from the $T^2$-dependence.

In the present magnetic insulating system, the thermal conductivity can be written as a sum of the phonon, magnon, and non-phononic quasiparticle contributions, $\kappa = \kappa_{ph} + \kappa_{mag} + \kappa_{qp}$. We first discuss $\kappa_{mag}$. Because the magnon gap is $\Delta_M/k_B \sim 2$ K except for the phase boundary regime, the magnon contribution is expected to become exponentially small in the temperature range shown in Fig. 8a–h. We note that regardless of whether the magnons are gapped or gapless, $\kappa_{mag}/T(T \to 0) = 0$. We next discuss the contribution of $\kappa_{ph}$. We point out that magnon–phonon scatterings do not play an important role in the

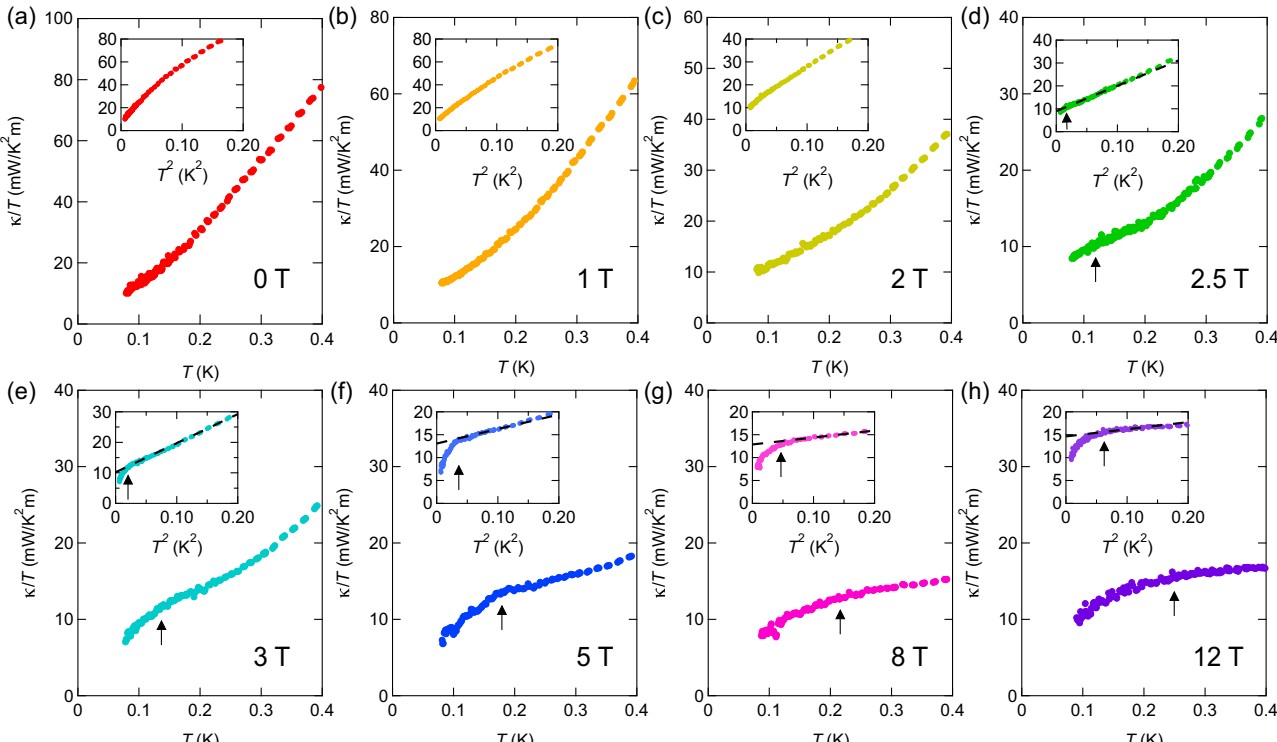

**Fig. 8 Thermal conductivity of YbIr$_3$Si$_7$. a–h** Thermal conductivity divided by temperature $\kappa/T$ of crystal #1 plotted as a function of $T$ in zero and magnetic field for $\mathbf{H}\,\|c$ at very low temperatures. The insets show $\kappa/T$ vs. $T^2$. The dashed straight lines in (**d–h**) represent the linear extrapolation from the high-temperature regimes. Arrows in the main panels and insets indicate the temperatures at which $\kappa/T$ deviates from the $T^2$-dependence.

phonon thermal conductivity because of the following reasons. If the magnon–phonon coupling are strong, the magnetic field would open up a gap in magnon spectrum, leading to the suppression of the magnon–phonon scattering, thus resulting in the enhancement of $\kappa_{ph}$ with $H$. However, as shown in Supplementary Fig. 3, $\kappa/T$ above 90 mK decreases monotonically with magnetic field up to 12 T. Moreover, $\kappa(H)$ changes little near the phase boundary between AF-I and AF-II, although the specific heat is distinctly enhanced by the magnetic fluctuations (Fig. 7a). As the phonon DOS does not change at the phase boundary, the observed field dependence indicates that phonons are little affected by magnons and non-phononic quasiparticles.

At low temperatures, $\kappa_{ph}$ is given by $\kappa_{ph} = \frac{1}{3}\beta_{ph}\langle v_s\rangle\ell_{ph}T^3$, where $\beta_{ph}$ is the phonon specific-heat coefficient obtained by the Debye phonon specific heat $C_{ph} = \beta_{ph}T^3$, $\langle v_s\rangle$ is the acoustic phonon velocity, and $\ell_{ph}$ is the effective mean free path of acoustic phonons. When $\ell_{ph}$ becomes comparable to the crystal size at very low temperatures (boundary limit), $\ell_{ph}$ is approximately limited by the effective diameter of the crystal $d_{eff} = \frac{2}{\pi}\left(\int_0^\alpha \frac{t}{\cos\theta}d\theta + \int_\alpha^{\pi/2}\frac{w}{\sin\theta}d\theta\right)$, where $w$ and $t$ are the width and thickness of the crystal, respectively, and $\alpha = \arctan(w/t)$. Using $\beta_{ph} = 0.45$ mJ mol$^{-1}$K$^4$ for LuIr$_3$Si$_7$, we estimate $\langle v_s\rangle \approx 3000$ m s$^{-1}$. We then find that $\ell_{ph}$ reaches the crystal size below 0.2 K. Using $d_{eff} = 0.17$ mm, we estimate the phonon thermal conductivity in the boundary limit $\kappa_{ph}^b/T \approx 4.9$ mW K$^{-2}$ m$^{-1}$ at the lowest temperature 0.08 K, which is smaller than the observed $\kappa/T$ for all fields. As $\kappa_{ph}^b$ gives the upper limit of the phonon thermal conductivity, this indicates that the thermal conductivity is dominated by $\kappa_{qp}$ in the low-temperature regime. Thus, the temperature and field dependencies of the thermal conductivity are mainly determined by the non-phononic quasiparticle contributions.

In zero field, the linear extrapolation of $\kappa/T$ to $T = 0$ has almost a zero intercept as seen in $\kappa/T$ vs. $T$ plot (main panel of Fig. 8a). On the other hand, the extrapolation to $T = 0$ has finite intercepts for both $\kappa/T$ vs. $T$ and $\kappa/T$ vs. $T^2$ plots for $\mu_0 H = 1$ and 2 T (Fig. 8b, c). This indicates that the quasiparticle thermal conductivity contains a finite residual $T$-linear term, $K_0 = \kappa_{qp}/T(T \to 0)$. We note that in the AF-I phase, similar magnitude of $K_0$, including vanishingly small $K_0$ in zero field, is observed in crystal #2. Moreover, in crystal #2 with similar effective diameter as #1, the magnitude of $\kappa/T$ at finite temperature is close to that of #1, suggesting similar mean free paths of phonon and quasiparticle in the AF-I phase (Supplementary Fig. S4). These results demonstrate the presence of mobile and gapless fermionic excitations in the AF-I phase. We stress that the observed finite $K_0$ does not originate from charged quasiparticles, in contrast to conventional metals. Evidence for this is provided by the spectacular violation of the Wiedemann-Franz (WF) law, which connects the electronic thermal conductivity $\kappa^e$ to the electrical resistivity $\rho$. In metals at low temperatures, the ratio $L = \kappa^e\rho/T \le L_0$ is satisfied, where $L_0 = (\pi^2/3)(k_B/e)^2 = 2.44 \times 10^{-8}$ W$\Omega$ K$^{-2}$ is the Lorenz number. The values of $K_0\rho_0$, where $\rho_0$ is the residual resistivity, are found to be $\sim 2.6 \times 10^3 L_0$ and $\sim 3.5 \times 10^3 L_0$ at $\mu_0 H = 1$ and 2 T, respectively. Here we used $K_0 = 6.4$ and 8.6 mW K$^{-2}$ m$^{-1}$ at $\mu_0 H = 1$ and 2 T, respectively, and $\rho_0 = 0.99$ $\Omega$cm for both fields. It is highly unlikely that the surface metallic region significantly violates the WF law. In fact, it is well known that the WF law holds in the 2D metals, even in the quantum Hall regime. We also note that the WF expectation of $L_0/\rho_0$ from the metallic surface is less than $2.5 \times 10^{-3}$ mW K$^{-2}$ m$^{-1}$, which is by far smaller than the experimental resolution. These results lead us to conclude that the neutral fermions in the insulating bulk state are responsible for the observed finite $K_0$. This suggests that, as the bulk resistivity diverges as $T \to 0$, the Lorenz number for the heat-carrying

quasiparticles also diverges. We also stress that the finite $K_0$ cannot be explained by the magnon excitations, as mentioned above. Thus, the thermal conductivity and specific-heat data under magnetic fields in the AF-I phase of YbIr$_3$Si$_7$ provide evidence for the presence of highly mobile and gapless neutral fermion excitations, which has been similarly reported in YbB$_{12}$.

We note parenthetically that finite values of both $\gamma$ and $K_0$ in the insulating states have been reported in quantum-spin-liquid candidates with 2D triangular lattices, including the organic compounds, EtMe$_3$Sb[Pd(dmit)$_2$]$_2$[29] and $\kappa$-H$_3$(Cat-EDT-TTF)$_2$[30], and inorganic compounds, 1$T$-TaS$_2$[31] and Na$_2$BaCo(PO$_4$)$_2$[32]. In EtMe$_3$Sb[Pd(dmit)$_2$]$_2$, although the presence or absence of finite $K_0$ has been controversial among different research groups[33,34], it has been shown very recently that differences between data sets are likely to be due to the cooling rate[35]. In 1$T$-TaS$_2$, as finite $K_0$ readily disappears by the introduction of disorder/impurity, the magnitude of $K_0$ appears to depend strongly on the sample quality[31]. These results suggest that high-quality single crystals are required to observe the finite $K_0$ in quantum spin-liquid systems. In the above compounds, finite $\gamma$ and $K_0$ have been discussed in terms of electrically neutral spinons forming the Fermi surface.

In the AF-II phase of YbIr$_3$Si$_7$, the magnitude of $\kappa/T$ is strongly reduced above 200 mK compared to that in the AF-I phase. As $\gamma$ in the AF-II phase is close to that of the AF-I phase except at the phase boundary, quasiparticle DOS is not largely different between two phases. Therefore, the suppression of $\kappa/T$ above 200 mK in the AF-II phase suggests that scattering time of neutral quasiparticles strongly depends on the magnetic structure. Moreover, a remarkable deviations from the $T^2$-dependence and suppression of $\kappa/T$ at very low temperatures are clearly observed. In the temperature regime where $\kappa/T$ is suppressed, $\kappa/T$ depends on $T$ as $\kappa/T \sim T^q$ with $q < 1$, which cannot be explained by a phonon contribution. Thus, the suppression of $\kappa/T$ indicates an opening of a tiny gap in the spectrum of the itinerant quasiparticle excitations. As this gap formation occurs below $\sim 0.3$ K, the estimate of the gap is two orders of magnitude smaller than the Kondo gap ($\sim 60$ K). We point out that there are two possible explanations for this behavior at very low temperatures well below $\sim 0.3$ K. One is a fully gapped thermal insulating state and the other is thermal semimetallic or nodal metallic state with a linearly vanishing DOS, as indicated by red shaded regime in Fig. 4. To clarify which scenario is realized, future measurements at lower temperature are required.

In the AF-II phase, $T^2$-dependent $\kappa/T$ in the high-temperature regime followed by a sharp drop at low temperatures is reproduced in a different crystal (#2, Supplementary Fig. 4). On the other hand, the magnitude of $\kappa/T$ of crystal #2 is largely enhanced compared to that of crystal #1. Moreover, the temperature at which the quasiparticle gap is formed in crystal #2 is nearly two times larger than that in crystal #1. As the $\gamma$-value is very close in both crystals in the AF-II phase, this difference is attributed to the mean free path of the neutral fermions.

We note that the specific heat measurements cannot resolve this gap formation due to the Schottky anomaly. The quasiparticle thermal conductivity $\kappa_{qp}$ is written as $\kappa_{qp}/T = K_0 + f(T)$. We find that the $T$-dependent part of $\kappa/T$, $f(T) + \kappa_{ph}/T$, can be fitted by a power-law dependence on $T$, as depicted in the insets of Supplementary Fig. 5a–h. The detailed $T$-dependence of $\kappa_{qp}$ is difficult to determine due to the presence of a small but finite $\kappa_{ph}/T$. The filled red circles in Fig. 7b represent $K_0$ obtained by extrapolating $\kappa/T$ to $T = 0$ using the power-law fits in the AF-I phase. After the initial rapid increase, $K_0$ increases slowly. In the AF-II phase, the quasiparticle contribution before the gap formation is obtained by the extrapolation from the high-temperature regime to $T = 0$. The filled red squares in Fig. 7b

show this quasiparticle contribution. For comparison, we also plot $\kappa/T$ at 90 mK. Interestingly, $\kappa/T$ obtained by high-temperature extrapolation appears to lie on top of the extrapolation from the AF-I phase. This suggests that $K_0$ steadily increases with magnetic field, but is strongly affected by the spin-flop transition. This field dependence indicates that the itinerant neutral fermions couple to the magnetic field and are strongly influenced by the magnetic ordering.

## Discussion

The combined results of the specific heat and thermal conductivity provide pivotal information on the neutral fermions observed in insulating materials. As shown by Fig. 7a, b, $\gamma$ and $K_0$ exhibit very different $H$-dependence. In particular, at zero field, while $\gamma$ is finite, no sizable $K_0$ is observed. We note that the result at zero field bears a resemblance to that of $SmB_6$. On the other hand, finite $\gamma$ and $K_0$ values in $YbB_{12}$ are similar to those of $YbIr_3Si_7$ in a finite field in the AF-I phase, although there is no signature of the phase transition at low field, as shown in Fig. 3a, b. In $YbB_{12}$, the $\gamma$ value is nearly sample independent, while $K_0$ values are strongly sample dependent, which is attributed to the amount of the impurities/defects determining the mean free path of the quasi-particles. In contrast, in the present study, we find strong field dependence of $K_0$ in a single sample, which cannot be due to the change in the impurity scattering. The rapid enhancement of $K_0 = \frac{1}{3}\gamma v \ell$, where $v$ and $\ell$ are velocity and mean free path of the neutral fermions, respectively, at low $H$ is attributed either to the increase of $\gamma$ or to the increase of $\ell$. As $\gamma$ is nearly constant in the AF-I phase except in the vicinity of the phase boundary, the enhancement of $K_0$ is attributed to the enhancement of $\ell$ of the quasiparticles. On the other hand, the absence of an enhancement of $K_0$ near the avoided QCP, despite the enhancement of $\gamma$, may be because $K_0$ is proportional to $\gamma\tau$, where $\tau$ is the scattering time. To explain why $K_0$ is not seriously affected by the enhancement of $\gamma$ near the QCP, it is required that $\tau$ is inversely proportional to the DOS of the neutral fermions, $\tau \propto 1/\gamma$. Such a mechanism is, for example, observed in $d$-wave superconducting materials, which show a universal residual thermal conductivity[36].

The nature and behavior of the novel charge-neutral fermions are not well understood; there are very few experimental results that can be used as tests of the various theoretical models, which include 3D Majorana fermions, composite magnetoexcitons, and spinons in fractionalized Fermi liquids. In this sense, our observations that the itinerant neutral fermions are very sensitive to the magnetic ordering can put significant restrictions on the various theories. The tiny gap formation (or a linearly vanishing DOS) in the AF-II phase indicates a transition from a thermal metal into an insulator (or a thermal semimetal), while the material remains an electrical insulator. This result demonstrates that the Fermi surface of the charge-neutral fermions becomes unstable towards gap formation at low temperatures, which is driven by the magnetic transition of the insulator. Therefore it is natural to consider that the neutral fermions are composed of strongly magnetically coupled $c$ and $f$ electrons through the Kondo effect. In this situation, neutral fermion excitations will be affected by AFM order and fluctuations. As revealed by the thermal conductivity measurements in two crystals, the scattering time of the charge-neutral fermions is largely different only in the AF-II phase. It is an open question why such a difference is present only in the thermally insulating/semiconducting phase but is absent in the thermally metallic phase. The clarification of the $f(T)$ term in $\kappa_{qp}$ would be key for understanding this remarkable difference between the AF-I and AF-II phases and is also an important future issue to understand the coupling between the neutral fermions and spin degrees of freedom.

In summary, we performed specific heat, thermal conductivity, and NMR measurements of bulk insulating $YbIr_3Si_7$ at low temperatures. In the low-field AF-I phase, we find finite $\gamma$ and $K_0$, demonstrating the emergence of itinerant gapless excitations even in the magnetic ground state of a Kondo insulator. A spectacular violation of the WF law directly indicates that $YbIr_3Si_7$ is a charge insulator but a thermal metal. More precisely, the charge-neutral quasiparticle excitations are either gapless or gapped with an extremely small excitation energy gap, much smaller than the base temperature 90 mK of our thermal conductivity measurements. A spin-flop transition at $\mu_0H \sim 2.5$ T is revealed by NMR measurements. With approaching the spin-flop transition, $\gamma$ is largely enhanced. Remarkably, inside the high-field AF-II phase, $\kappa/T$ exhibits a sharp drop at very low temperatures, indicating the opening of a tiny gap much smaller than the Kondo gap or a linear vanishing DOS of the neutral excitations. This demonstrates a field-induced transition from a thermal metal into an insulator/semimetal driven by the spin-flop transition. The present results demonstrate that the neutral fermions are directly coupled to the spin degrees of freedom, which have never been considered in existing theories. Our experimental observations impose a strong constraint on the theories of charge-neutral fermions. Thus, $YbIr_3Si_7$ provides an intriguing platform for studying the neutral fermions in strongly correlated insulators.

## Methods

**Crystal growth and resistivity**. Single crystals of $YbIr_3Si_7$ have been grown using the laser pedestal technique. The crystals #1, #3, and #4 were taken from the same batch and crystal #2 was taken from the different batch. The crystals were cut from the as-grown ingot and polished into a rectangular shape, with the longest direction corresponding to the $a$ axis and the shortest direction to the $c$ axis. Back scattering X-ray Laue diffraction was used to orient the crystals. The dimension of the samples used for transport and heat capacity measurements are $2.7 \times 1.6 \times 0.10$ mm³ (#1) and $2.3 \times 0.66 \times 0.3$ mm³ (#2). Resistivity were measured with a.c. technique in a standard four-contact configuration. The same contacts were used for the thermal conductivity measurements.

**Specific heat**. The specific heat of $YbIr_3Si_7$ single crystals was measured by a long-relaxation calorimetry using a bare chip Cernox sensor, which is used as a thermo-meter and a heater. The specific heat of the addenda including the grease was measured before the sample was mounted. The specific heat of the sample is obtained by subtracting the addenda from the total specific heat measured with the sample.

**Nuclear magnetic resonance**. $^{29}Si$-nuclear magnetic resonance (NMR) mea-surements were performed in applied magnetic field parallel to the $c$ axis. A conventional spin-echo technique was used.

**Thermal conductivity**. The thermal conductivity were measured by the steady-state method, applying the current **q** along the $a$ axis. Magnetic field was applied along the $c$ axis. The thermal gradients $\nabla T$ were detected by $RuO_2$ thermometers, and $\kappa$ were obtained as $\kappa = q/\nabla T$. The $RuO_2$ thermometers were calibrated by a commercial $RuO_2$ thermometer (LakeShore) with zero-field calibrations. The field dependence of the $RuO_2$ thermometers was calibrated by using a Coulomb blockade thermometer, which is insensitive to the magnetic fields. The validity of the calibration of the $RuO_2$ thermometers was carefully checked by confirming the WF law of a thin gold wire in magnetic fields in the same setup.

**Discussions of thermal decoupling and thermal leakage**. (1) Decoupling: In metals, thermal decoupling of phonons and electrons caused by the poor contact leads to the anomalous rapid suppression in the thermal conductivity at low temperatures, as reported in cuprate superconductors[37]. To ensure good thermal contacts, we sputtered gold on fresh surface of the crystal and then attached the contacts with silver epoxy. At room temperature and at ~100 K, the contact resistance is much less than 1 Ω. While the downturn of the thermal conductivity with decreasing $T$ is observed only in the AF-II phase above 2.5 T, it is absent in the AF-I phase at lower fields. The fact that the finite residual temperature-linear term in the thermal conductivity is observed down to the lowest temperature in the AF-I phase provides evidence that quasiparticles are well coupled to phonons, i.e., the absence of thermal decoupling. (2) Thermal leakage: We measured the thermal conductivity of thin stainless wire, whose thermal resistance is about two orders of magnitude larger than the present crystal, in the same setup. We confirmed the Wiedemann-Franz law in the stainless wire, demonstrating negligibly small thermal leakage.

## Data availability

The data that support the findings of this study are available from the corresponding author upon reasonable request.

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

## Acknowledgements

A.H.N. and E.M. acknowledge fruitful discussions with Chris Hooley. Y.M. acknowledges discussion with H. Kontani, Lu Li, and J. Singleton. L.Q., and E.M. acknowledge support from the U.S. Department of Energy for Grant No. DE-SC0019503. A.H.N. was supported by the National Science Foundation grant No. DMR-1917511 and the Robert A. Welch Foundation grant C-1818. This work is supported by Grants-in-Aid for Scientific Research (KAKENHI) (Nos. JP15H02106, JP18H01177, JP18H01178, JP18H01180, JP18H05227, JP19H00649, JP20H02600, JP18K03511, and JP20H05159) and on Innovative Areas "Quantum Liquid Crystals" (No. JP19H05824) from Japan Society for the Promotion of Science (JSPS), and JST CREST (JP-MJCR19T5).

## Author contributions

L.Q. and E.M. grew the high-quality single-crystalline samples. Y.S., T.T., Y.K., and S. Kasahara performed resistivity, magnetization, specific heat, and thermal conductivity measurements. T.K., S. Kitagawa, and K.I. performed nuclear magnetic resonance measurements. Y.S., T.T., Y.K., and S. Kasahara analyzed the data. Y.S., S.S., Y.K., R.P., T.S., A.H.N., E.M., and Y.M. discussed and interpreted the results. Y.S., Y.K., R.P., T.S., A.H.N., and Y.M. prepared the manuscript, with input from all other co-authors.

## Competing interests

The authors declare no competing interests.
