## [Peer Review File. · Nature Communications]

REVIEWER COMMENTS

Reviewer #1 (Remarks to the Author):

The authors present an experimental study combining low temperature thermal conductivity, specific heat and magnetisation measurements to draw conclusion about the behaviour of fermionic excitations in the purported Kondo lattice material SrIr₃Si₇.

The main findings are that i) magnetisation and NMR experiments suggests an AFM to AFM spin flop transition at low fields and temperatures and ii) an extrapolation of κ/T as $T \rightarrow 0$ is finite for $B > 0T$, in a regime where the electrical resistivity also diverges. From these observations the authors conclude the presence of gapless excitations that are field-induced, which are interpreted in terms of neutral fermions.

I agree with the authors that this is an interesting area of physics with some significant unresolved problems. In particular the existence of quantum oscillations in the insulating state of SmB₆ poses many fundamental theoretical questions, and studies of heat capacity and thermal transport in similar systems are very desirable.

While the subject is interesting, and the techniques are used in an appropriate way, I believe that there is further work to be done to rule out more prosaic explanations for the data. I outline my concerns below.

1. Thermal transport measurements at $T < 1K$ are subject to numerous potential sources of systematic error. Firstly, heat losses resulting from thermally poor contacts can cause an overestimate in the value of κ , and if the contacts are particularly poor, can cause a downturn in a plot of κ/T vs T [1]. What evidence do the authors have that their contacts are sufficiently thermally conductive at the temperatures measured to avoid these problems? Specifically, is the thermal conductance of the sample plus contacts much greater than that of the measurement leads?

2. Resistive RuO₂ thermometers have an R vs T curve that shifts with magnetic field. What steps have the authors taken to reliably calibrate their thermometers at low fields and in an applied magnetic field, and can a miscalibration be ruled out as the cause of the finite linear term appearing at $B > 0$?

3. The authors use a power law method to extrapolate their thermal conductivity data to $T=0$ (insets of figures 8a) – 8h)). This is key to the establishment of κ_0/T and underpins the main conclusions of the paper. Some of the power laws observed are very low, for instance $p=1.3$ for $B=0T$. This is never

observed as arising from phonons in an insulating system (e.g. sapphire). What further reassurance can the authors give that these samples are in the boundary scattering regime, where such a power law extrapolation is reliable? Using the measured values of heat capacity and an appropriately calculated (or measured) sound velocity, is the expected phonon mean free path of the same order as the sample dimensions?

4. Related to this, for fields of 3T and above a downturn in the κ/T data develops. Why are extrapolations at these fields performed using data at temperatures higher than the downturn temperature?

5. Leaving aside the issue of extrapolations, the data has some inconsistencies with the interpretation offered. Comparing the κ/T curves for the various fields in figure 8, one sees that the effect of a field at finite temperatures (say 400mK) is to suppress the thermal conductivity from 80 mW/K²cm at 0 T to 15 mW/K²cm at 12 T. If the sample is moving from an insulating state to a thermal metal, why should the thermal conductivity at 400mK be suppressed in this way? A possible explanation is that the induced heat carriers are scattering phonons, but it follows that the sample is not in the boundary limited scattering regime, which makes a power law extrapolation incorrect.

As it stands, due to the concerns above, I would not recommend publication of the manuscript until further work is done to reassure the reader on these.

[1] See for instance Smith et al, Phys. Rev. B, 71, 014506 (2005)

Reviewer #2 (Remarks to the Author):

This work contains a series of interesting results on various properties of the heavy fermion compound YbIr₃Si₇. The results are very interesting, but I have several outstanding concerns about the study:

1) My main concern with this study is that the authors seems to have completely ignored the important point of studying the sample dependence of their results. They are making bold claims about exotic states of matter. Such bold claims should be met with higher standards of investigation of sample-to-sample variability of their findings. Apart from the measurement of resistivity that shows two samples, I could not find any comment on which samples were used for any of the other measurements. Are the NMR, specific heat, heat conductivity on the same and single sample? is this one of the samples for the resistivity? which one?

There are certainly very strange and interesting behaviors being reported, but one would like to see how these change depending on sample quality. I am inclined to not recommend this paper for publication unless the authors report on sample variability at least for specific heat and the heat conductivity measurements (which are central to the claims they want to advance).

2) What is the evidence that the state AFM-II is actually an antiferromagnet? just the fact that the NMR peak does not shift substantially from the position at higher temperatures?

3) In the insets of Fig.5 it is not clear which atoms are in the same planes (namely which ones have a common projection of their position to the c axis). Can the authors add a sketch of these planes.

4) I find the specific heat data strange and its explanations not very convincing. First, why is there apparently no contribution of phonons to the specific heat data? Phonons should contribute with a $C \sim T^3$. Therefore, after subtracting the Schottky anomaly, they should contribute as a straight line when plotting C/T vs T^2 . The plots in Fig.6 don't show this. Instead they show a contribution with a power larger than T^3 , almost looks like a $C \sim T^4$. The authors barely comment on this. But never try to explain why there are apparently no phonons in their measurement?

5) In relation to the above the authors write:

"As seen in Fig. 2(b), C/T increases steeper than $C/T \propto T^2$ line in the whole field range, indicating that α is larger than 2. The AFM spin-wave theory predicts $\alpha = 1$ and 2 for quasi-2D and 3D systems, respectively. Therefore, the results demonstrate the presence of contributions other than spin waves."

This is a strange statement. Gapless spin waves should contribute to the specific heat as a power law. But when the spin waves are gapped their contribution should be exponentially suppressed below the gap. The Zeeman energy scale at $B=1T$ for a spin with moment 1.5 Bohr magnetons (as they write for the Yb ions), is $\sim 1K$. So even ignoring the pinning scale of spin orbit coupling, we expect a Zeeman gap of about 1K for the spin waves at 1T. So it is not clear what is the relation of the spin waves to the low temperature specific heat they are studying.

6) The authors write in the discussion section:

"The rapid enhancement of κ_0/T at low H is attributed to the increase of the number of itinerant heat carriers and/or the change of the dispersion of the neutral particles, which results in the increase of the group velocity and hence the mean free path. As γ is nearly constant in the AF-I phase, the enhancement of κ_0/T is attributed to the increase of the mean free path."

The two adjacent statements above read almost in contradiction to each other.

7) What is the authors explanation for the power law of the κ/T in AFM-I changing continuously with in-plane field?

8) The authors write:

"the difference of the saturation values of ρ between crystals #1 ($\sim 1 \Omega\text{cm}$) and #2 ($\sim 2.1 \Omega\text{cm}$) can be quantitatively explained by the area and thickness of the crystal planes used for the measurements"

However, I could not find any explicit quantitative estimation of the resistance of their samples to back up the statement.

9) The authors write: "Similar phenomena have been reported in SmB6 and

YbB12, in which the metallic conductivity takes place at the surfaces of the crystal, while electronic transport and optical measurements clearly show the opening of a finite charge gap in the bulk at low temperatures."

What study do the authors have in mind when referring to an optical gap in these materials? In the case of SmB_6, there is quite substantial optical conductivity inside the gap, and actually not a very clear optical at all (see Phys. Rev. B 94, 165154 (2016)).

10) The authors write: "The enhancement of the specific heat above T_N might indicate the importance of short-range order."

It is not clear which enhancement they are trying to highlight. It is not clear what they mean by "short-range order" and why would it lead to an enhancement of the specific heat.

11) The authors write: "Our experimental observations impose a strong constraint on the theories of charge-neutral fermions."

What are these strong constraints?

-- --

Also there appears to be a typo when they write:

"We find that κ/T for $\mu_0 H=0, 1, \text{ and } 2T$ is well fitted as $\kappa/T=\kappa_0/T+c_1 T^p$."

this should probably be " $\kappa/T=c_0+c_1 T^p$ ", otherwise it will imply a finite heat conductivity at zero temperature which is not thermodynamically sensible (and not a finite κ/T).

Reviewer #1 (Remarks to the Author):

The authors present an experimental study combining low temperature thermal conductivity, specific heat and magnetisation measurements to draw conclusion about the behaviour of fermionic excitations in the purported Kondo lattice material SrIr_3Si_7 . The main findings are that i) magnetisation and NMR experiments suggests an AFM to AFM spin flop transition at low fields and temperatures and ii) an extrapolation of κ/T as $T \rightarrow 0$ is finite for $B > 0\text{T}$, in a regime where the electrical resistivity also diverges. From these observations the authors conclude the presence of gapless excitations that are field-induced, which are interpreted in terms of neutral fermions.

I agree with the authors that this is an interesting area of physics with some significant unresolved problems. In particular the existence of quantum oscillations in the insulating state of SmB_6 poses many fundamental theoretical questions, and studies of heat capacity and thermal transport in similar systems are very desirable.

While the subject is interesting, and the techniques are used in an appropriate way, I believe that there is further work to be done to rule out more prosaic explanations for the data. I outline my concerns below.

Response:

We thank Reviewer #1 for careful reading of our manuscript and insightful comments, which helped us improve our manuscript. We are pleased that Reviewer #1 finds our research subject interesting.

1. Thermal transport measurements at $T < 1\text{K}$ are subject to numerous potential sources of systematic error. Firstly, heat losses resulting from thermally poor contacts can cause an overestimate in the value of κ , and if the contacts are particularly poor, can cause a downturn in a plot of κ/T vs T [1]. What evidence do the authors have that their contacts are sufficiently thermally conductive at the temperatures measured to avoid these problems? Specifically, is the thermal

conductance of the sample plus contacts much greater than that of the measurement leads?

Response:

Reviewer #1 raises questions concerning decoupling, contact resistance, and thermal leakage.

Decoupling: In metals, thermal decoupling of phonons and electrons caused by poor contact often leads to the rapid suppression in the thermal conductivity at low temperatures, as reported in cuprates. Similar decoupling phenomena between quasiparticles and phonons are also expected due to poor contact in the present experiments. We therefore have paid careful attention to the contacts. To ensure good thermal contacts, we sputtered gold on the fresh surface of the crystal and then attached the contacts with silver epoxy. At room temperature and at ~ 100 K, we confirmed that the contact resistance is much less than 1Ω . Unfortunately, it is difficult to measure the contact resistance at very low temperatures because of the insulating resistivity. However, we stress that while the downturn of the thermal conductivity with decreasing T is observed in the AF-II phase above 2.5 T, it is absent in the AF-I phase at lower fields. According to M. F. Smith *et al.* (Phys. Rev. B 71, 014506 (2005)), the downturn behavior is more pronounced with increasing the κ/T values at low T . On the other hand, in the AF-II phase of YbIr_3Si_7 , κ/T at 90 mK rather decreases with increasing H (see Fig. 7(b)). Thus, the downturn behavior observed only in the AF-II phase cannot be explained by the thermal decoupling. The fact that the metallic-like temperature-linear thermal conductivity is observed down to the lowest temperature in the AF-I phase also supports that quasiparticles are well coupled to phonons, i.e., the absence of thermal decoupling.

Thermal leakage: We measured the thermal conductivity of thin stainless wire in the same setup used for the present measurements. The contact resistance for the stainless wire is comparable to that for the present crystal at room temperature and at 100 K. The total thermal resistance of the stainless wire including the contact resistance is one-two orders of magnitude larger than that of the present crystal at 0.1 K. We then confirmed the Wiedemann-Franz law in this wire, demonstrating that thermal conductivity is precisely measured

irrespective of the presence of the contact resistance and that thermal leakage is negligibly small during the measurements.

We described the above results in the method section of the revised manuscript.

2. Resistive RuO₂ thermometers have an R vs T curve that shifts with magnetic field. What steps have the authors taken to reliably calibrate their thermometers at low fields and in an applied magnetic field, and can a miscalibration be ruled out as the cause of the finite linear term appearing at B>0?

Response:

We use a commercial RuO₂ thermometer (LakeShore) with zero-field calibrations to calibrate the RuO₂ thermometers used for the thermal conductivity measurements at 0 T. The field dependence of the RuO₂ thermometers was calibrated (0.25 T step below 2T and 0.5T step above 2T) in our laboratory by using a Coulomb blockade thermometer, which is insensitive to the magnetic fields. The validity of the calibration of the RuO₂ thermometers is carefully checked by confirming the Wiedemann-Franz law of thin Au wire in magnetic fields in the same setup.

3. The authors use a power law method to extrapolate their thermal conductivity data to T=0 (insets of figures 8a) – 8h)). This is key to the establishment of $\kappa_{0/T}$ and underpins the main conclusions of the paper. Some of the power laws observed are very low, for instance $p=1.3$ for B=0T. This is never observed as arising from phonons in an insulating system (e.g. sapphire). What further reassurance can the authors give that these samples are in the boundary scattering regime, where such a power law extrapolation is reliable? Using the measured values of heat capacity and an appropriately calculated (or measured) sound velocity, is the expected phonon mean free path of the same order as the sample dimensions?

Response:

Although we mentioned that the phonon thermal conductivity κ_{ph} is in the boundary limit at 0.1K, the system is not in the boundary limit at 0.2 K because the phonon mean free path estimated from the specific heat is still less than the crystal size. In the boundary-limited regime, we expect $p = 2$ for the diffusive scattering limit and $p = 1$ for the specular reflection limit. In real systems, p often takes an intermediate value ($1 < p < 2$) as reported in various compounds, such as Al_2O_3 , Si, KCl, KBr, LiF, diamond, and Nd_2CuO_4 , (S. Y. Li *et al.*, Phys. Rev. B **77**, 134501 (2008)). Although phonons might explain the power-law in the boundary-limited regime, the finite offset cannot be explained by phonons. Thus, the conclusion of the finite κ/T in the zero-temperature limit, which is the most important point in the current manuscript, does not change because such a phonon contribution cannot explain it, and we need to conclude the existence of charge-neutral quasiparticles.

We amended the sentence as follows: “Using $d_{\text{eff}} = 0.17$ mm, we estimate the phonon thermal conductivity in the boundary limit $\kappa_{\text{ph}}^{\text{b}}/T \approx 4.9$ mW/K²m at the lowest temperature 0.08 K, which is smaller than the observed κ/T for all fields.” We note that this change does not affect the conclusion.

4. Related to this, for fields of 3T and above a downturn in the κ/T data develops. Why are extrapolations at these fields performed using data at temperatures higher than the downturn temperature?

Response:

In the low-temperature regime where a downturn behavior of κ/T is observed, κ/T depends on the temperature as T^p with $p < 1$. Therefore, this downturn behavior has a non-phononic origin, and hence it is caused by the gap formation in the itinerant quasiparticle excitations. On the other hand, κ/T is well fitted by T^p with $p > 1$ in the high-temperature regime, similar to that in the AF-I phase at low fields. Therefore, to obtain the quasiparticle contribution before the gap formation, we extrapolate κ/T from the high-temperature regime. We explicitly stated this in the revised manuscript.

5. Leaving aside the issue of extrapolations, the data has some inconsistencies with the interpretation offered. Comparing the κ/T curves for the various fields in figure 8, one sees that the effect of a field at finite temperatures (say 400mK) is to suppress the thermal conductivity from 80 mW/K²cm at 0 T to 15 mW/K²cm at 12 T. If the sample is moving from an insulating state to a thermal metal, why should the thermal conductivity at 400mK be suppressed in this way? A possible explanation is that the induced heat carriers are scattering phonons, but it follows that the sample is not in the boundary limited scattering regime, which makes a power law extrapolation incorrect.

Response:

The sample does not change from an insulating state to a thermal metal but from a thermal metal to an insulating state as the field is increased. As shown in Supplementary Fig. 3, κ/T shows little change as a function of H near the phase boundary between AF-I and AF-II. On the other hand, as shown in Fig. 7(a), γ and C/T are strongly enhanced near the boundary due to the critical magnetic fluctuations. These results provide evidence that quasiparticles are rarely scattered by phonons.

In the revised manuscript, we discuss the magnon contributions to the specific heat and thermal conductivity. In the temperature regime where the thermal conductivity is discussed, the magnon contribution is vanishingly small because the temperature is well below the magnon gap.

As the γ -value in the AF-II phase is similar to that of the AF-I phase except at the phase boundary, the quasiparticle density of states is not largely different between the two phases. Therefore, the result that κ/T in the AF-I phase is much larger than that in the AF-II phase at 400 mK implies that the scattering time of the neutral quasiparticles strongly depends on the magnetic structure. We explicitly state this in the revised manuscript.

As it stands, due to the concerns above, I would not recommend publication of the manuscript until further work is done to reassure the reader on these.

[1] See for instance Smith et al, Phys. Rev. B, 71, 014506 (2005)

Response:

We believe we have addressed all the concerns raised by Reviewer #1.

Reviewer #2 (Remarks to the Author):

This work contains a series of interesting results on various properties of the heavy fermion compound YbIr_3Si_7 . The results are very interesting, but I have several outstanding concerns about the study:

Response:

We thank Reviewer #2 for carefully reviewing our paper and for the constructive comments that helped us to improve our manuscript. To address Reviewer #2's concerns, we have performed additional experiments and added discussions in the revised manuscript.

1) My main concern with this study is that the authors seems to have completely ignored the important point of studying the sample dependence of their results. They are making bold claims about exotic states of matter. Such bold claims should be met with higher standards of investigation of sample-to-sample variability of their findings. Apart from the measurement of resistivity that shows two samples, I could not find any comment on which samples were used for any of the other measurements. Are the NMR, specific heat, heat conductivity on the same and single sample? is this one of the samples for the resistivity? which one?

There are certainly very strange and interesting behaviors being reported, but one would like to see how these change depending on sample quality. I am inclined to not recommend this paper for publication unless the authors report on sample variability at least for specific heat and the heat conductivity measurements (which are central to the claims they want to advance).

Response:

As mentioned in the manuscript (p.5, subsection II-B, the first paragraph; p.9, subsection III-B, the second paragraph), the specific heat and thermal conductivity are measured in crystal #1. In the NMR experiments, a different crystal taken from the same batch is used, because a large sample volume is needed. We have added this information in the revised manuscript.

As pointed out by Reviewer #2, the sample variability is crucially important. To address this issue, we have performed specific heat and thermal conductivity measurements on crystal #2 grown in a different batch.

As shown in Supplementary Fig. 2(a) and (b), C/T data of crystal #2 at 0 T and 12 T well coincides with those of crystal #1 in the whole temperature range. Thus specific heat is sample independent and one of the main conclusions, i.e. finite γ term, is reproduced.

Supplementary Fig. 4(a)-4(g) show κ/T of #2 in zero and finite magnetic fields for $\mathbf{H} \parallel c$ at very low temperatures. The size of #2 is 2.3 (l : length) x 0.66 (w : width) x 0.30 (t : thickness) mm³, and that of #1 is 2.7x1.6x0.10mm³. Then the effective diameter of #2 is comparable to #1.

In the AF-I phase, the magnitude of κ/T and its temperature dependence in crystal #2 are close to those of #1. Thus, one of the main conclusions, i.e., thermal metallic state in the AF-I phase, is reproduced.

In the AF-II phase, we found that the T^2 -dependent κ/T in the high-temperature regime followed by a sharp drop at low temperatures observed in crystal #1 is also observed in #2. Thus one of the main conclusions, i.e. thermal insulating state in the AFM-II phase, is reproduced. On the other hand, the magnitude of κ/T of #2 is largely enhanced compared to that of #1. Moreover, the temperature at which the gap is formed in #2 is nearly two times larger than that in #1. As the γ -value of #2 is very close to that of #1 even in the AF-II phase, this difference is attributed to the difference of the mean free path of the quasiparticles. The instability of the emergent Fermi surface of neutral fermion may be sensitive to the mean free path, similar to the case of disorder-sensitive pairing instability of electrons in anisotropic superconductors. These new results would stimulate systematic studies of the impurity effect on the gap opening

found in the present study, which may provide further information on this novel insulating state.

In the revised manuscript, we have added the above discussion on the sample dependence.

2) What is the evidence that the state AFM-II is actually an antiferromagnet? just the fact that the NMR peak does not shift substantially from the position at higher temperatures?

Response:

We concluded that the AF-II is an antiferromagnetic state not only from NMR but also from magnetization and magnetic field dependence of transition temperature. Above 2.5 T, at which the spin-flop transition occurs, the magnetization increases gradually with H without showing saturation. Moreover, as shown by the specific heat, a sharp phase transition is observed even above 2.5 T. In addition, the transition temperature slowly decreases with H . These results support that the AF-II is an antiferromagnetic state, inconsistent with a ferromagnetic state.

The NMR spectrum in the AF-I phase is nicely reproduced by a Γ_1 -antiferromagnetic state with $1.5\mu_B$, which was reported by neutron diffraction measurements. Using the same ordered moment, the NMR spectrum in the AF-II phase can be reproduced by a canted antiferromagnetic state as shown in the inset of Fig. 5(b) but cannot be reproduced by a ferromagnetic state.

In the revised manuscript, we added the above arguments.

3) In the insets of Fig.5 it is not clear which atoms are in the same planes (namely which ones have a common projection of their position to the c axis). Can the authors add a sketch of these planes.

Response:

We have revised the inset of Fig. 5.

4) I find the specific heat data strange and its explanations not very convincing. First, why is there apparently no contribution of phonons to the specific heat data? Phonons should contribute with a $C \sim T^3$. Therefore, after subtracting the Schottky anomaly, they should contribute as a straight line when plotting C/T vs T^2 . The plots in Fig.6 don't show this. Instead they show a contribution with a power larger than T^3 , almost looks like a $C \sim T^4$. The authors barely comment on this. But never try to explain why there are apparently no phonons in their measurement?

Response:

The phonon contribution of specific heat (C_{ph}) is estimated from the specific heat of isostructural LuIr_3Si_7 (see solid lines in Figs. 2(a) and (b)). As the Lu ion is nonmagnetic and its atomic number is next to Yb, this compound is a nonmagnetic reference of YbIr_3Si_7 . Below 1 K, the phonon contribution is less than 1% of the total specific heat. (In Fig. 6, C_{ph}/T overlaps with the bottom axis). In the manuscript, we have explicitly mentioned that the phonon contribution is negligibly small (p.7, subsection III-A, the first paragraph).

5) In relation to the above the authors write:

"As seen in Fig. 2(b), C/T increases steeper than $C/T \propto T^2$ line in the whole field range, indicating that α is larger than 2. The AFM spin-wave theory predicts $\alpha = 1$ and 2 for quasi-2D and 3D systems, respectively. Therefore, the results demonstrate the presence of contributions other than spin waves." This is a strange statement. Gapless spin waves should contribute to the specific heat as a power law. But when the spin waves are gapped their contribution should be exponentially suppressed below the gap. The Zeeman energy scale at $B=1\text{T}$ for a spin with moment 1.5 Bohr magnetons (as they write for the Yb ions), is $\sim 1\text{K}$. So even ignoring the pinning scale of spin orbit coupling, we expect a Zeeman gap of about 1K for the spin waves at 1T. So it is not clear what is the relation of the spin waves to the low temperature specific heat they are studying.

Response:

We appreciate this comment. We realized that the spin-wave contribution is important in the specific heat in the high-temperature regime.

There are two possibilities of the magnon contribution. The first is that the magnon is gapless and the Zeeman gap opens at a finite magnetic field, as pointed out by Reviewer #2. Second is that the spin-wave excitations are gapped even in zero field due to magnetic anisotropy. As shown in Fig. 6(a), C/T in zero field increases more steeply than $C/T \propto T^2$ line expected in gapless 3D magnon above 1 K. Thus, the gapless magnon cannot explain the present specific heat data. In contrast, we find that the total specific heat is well fitted by assuming the gapped magnon excitations as $C(T)/T = \gamma + C_{\text{Mag}}/T + C_{\text{Sch}}(T)/T$, where $C_{\text{Mag}} = \beta_{\text{M}} T^3 \exp(-\Delta_{\text{M}}/k_{\text{B}} T)$ (Δ_{M} is the magnon gap and β_{M} is a coefficient) is the magnon contribution. Magnons have a gap even in zero field because Yb^{3+} magnetic ion usually has a large magnetic anisotropy due to the strong LS coupling.

As shown in Figs. 6(b)-6(h), the temperature dependence of the specific heat in magnetic field is also well fitted by the above formula including gapped magnon. Supplementary Fig. 1 shows the field dependence of Δ_{M} . Δ_{M} is suppressed with approaching the phase boundary, and in the high-field regime of the AF-II phase, Δ_{M} is nearly independent of H . In the ordered phase, Δ_{M} is determined by the competition between the Zeeman field and the molecular (mean) field due to the magnetic moment around the magnetic ions. When magnetic order is stabilized, the molecular field dominates, and Δ_{M} is not seriously influenced by the Zeeman field. Because the magnetic order is suppressed and magnetic fluctuations are enhanced near the phase boundary, Δ_{M} is suppressed, consistent with the observed behavior of Δ_{M} . To observe the magnon gap directly, we need neutron scattering experiments, but this is beyond the scope of this manuscript. It is important to note that the finite γ cannot be explained by gapless or gapped magnons.

We stress that, because the magnon gap is $\Delta_{\text{M}}/k_{\text{B}} \sim 2$ K except for the phase boundary regime, the magnon contribution in κ/T is expected to be exponentially small in the temperature ranges we discussed the thermal conductivity.

Therefore, when continuing the data of the fitting to $T = 0$, the resulting finite κ/T at $T \rightarrow 0$ extrapolated from high temperatures cannot be explained by magnon but by fermions (Even if gapless magnon is present in this compound, κ_{mag}/T becomes zero in the zero-temperature limit). We explicitly state this in the revised manuscript.

6) The authors write in the discussion section:

"The rapid enhancement of κ_0/T at low H is attributed to the increase of the number of itinerant heat carriers and/or the change of the dispersion of the neutral particles, which results in the increase of the group velocity and hence the mean free path. As γ is nearly constant in the AF-I phase, the enhancement of κ_0/T is attributed to the increase of the mean free path."
The two adjacent statements above read almost in contradiction to each other.

Response:

To avoid confusion, we rephrased the sentences as follows:

The rapid enhancement of $K_0 = \gamma v l$ at low H , where v and l are velocity and mean free path of the neutral quasiparticle, respectively, is attributed either to the increase of γ or the increase l . As γ is nearly constant in the AF-I phase except in the vicinity of the phase boundary, the enhancement of K_0 is attributed to the enhancement of l of the quasiparticles.

7) What is the authors explanation for the power law of the kappa/T in AFM-I changing continuously with in-plane field?

Response:

In all measurements, the magnetic field is applied parallel to the c -axis (out-of-plane field). As γ is nearly constant in the AF-I phase and quasiparticle-phonon scattering is weak as discussed in the main text (see Supplementary Fig. 3 and Fig. 7(b) in the main text), the continuous change of the power of κ/T is attributed to the field dependence of the quasiparticle mean free path, possibly due to the change of the quasiparticle dispersion.

8) The authors write:

"the difference of the saturation values of ρ between crystals #1 ($\sim 1 \Omega\text{cm}$) and #2 ($\sim 2.1 \Omega\text{cm}$) can be quantitatively explained by the area and thickness of the crystal planes used for the measurements" However, I could not find any explicit quantitative estimation of the resistance of their samples to back up the statement.

Response:

The sample dimensions of #1 and #2 crystals are t (thickness) = 0.1 mm, w (width) = 1.63 mm, and l (contact distance) = 1.55 mm for #1, and $t = 0.3$ mm, $w = 0.66$ mm, $l = 1.03$ mm for #2, respectively. The saturation resistivity at the lowest temperature is 0.97 Ωcm and 2.3 Ωcm for #1 and #2, respectively. The ratio $\rho(\#2)/\rho(\#1) = 2.4$ is close to the ratio of the thickness $t(\#2)/t(\#1) = 3.0$. Thus, the saturation values of ρ roughly follow the relation of $\rho \propto t$ (t is the thickness of the crystal). The small deviation is due to the contribution from the side of the crystal. These results are consistent with the presence of the surface conduction channel, as reported in [23].

9) The authors write: "Similar phenomena have been reported in SmB₆ and YbB₁₂, in which the metallic conductivity takes place at the surfaces of the crystal, while electronic transport and optical measurements clearly show the opening of a finite charge gap in the bulk at low temperatures."

What study do the authors have in mind when referring to an optical gap in these materials? In the case of SmB₆, there is quite substantial optical conductivity inside the gap, and actually not a very clear optical at all (see Phys. Rev. B 94, 165154 (2016)).

Response:

Although a clear optical gap is not observed in SmB₆, a clear optical gap has been reported in YbB₁₂ (H. Okamura, *et al.*, J. Phys. Soc. Japan. 74, 1954 (2005)). In both SmB₆ and YbB₁₂, electrical transport measurements suggest the

opening of the charge gap in the bulk. In the revised manuscript, we eliminated the argument of optical gap and revised the sentences as follows:

Similar phenomena have been reported in SmB_6 and YbB_{12} , in which the metallic conductivity takes place at the surfaces of the crystal, while electronic transport suggests the opening of a finite charge gap in the bulk at low temperatures.

10) The authors write: "The enhancement of the specific heat above T_N might indicate the importance of short-range order."

It is not clear which enhancement they are trying to highlight. It is not clear what they mean by "short-range order" and why would it lead to an enhancement of the specific heat.

Response:

We tried to highlight that in Fig. 2(a), C/T increases on approaching the Néel temperature from above. To avoid confusion, we rephrase the sentence as follows:

As depicted in Fig. 2(a), C/T increases on approaching the Néel temperature from above. Similar phenomena have been observed in many antiferromagnets, such as CeRhIn_5 . This increase of the specific heat above T_N is attributed to the entropy release associated with the short-range antiferromagnetic order or fluctuations.

11) The authors write: "Our experimental observations impose a strong constraint on the theories of charge-neutral fermions." What are these strong constraints?

Response:

After the discoveries of quantum oscillations in Kondo insulators, many theoretical proposals have been put forward to explain this unusual phenomenon. Moreover, recently reported metallic thermal conduction have

revealed the presence of the unknown exotic neutral quasiparticles in Kondo insulators. Then the relationship between the quantum oscillations and neutral fermions has drawn much attention. The neutral fermions have been proposed in various theoretical models, including Majorana fermions, composite excitons, and spinons. However, there are no clues to the identity of the unknown neutral fermion, and its origin is largely elusive.

The present results provide several constraints on the theories of charge-neutral fermions in Kondo insulators. First, the charge-neutral fermions emerge even in the magnetic ground state. This is a piece of completely new information, because the charge-neutral fermions in Kondo insulators have been discussed in non-magnetic compounds, SmB_6 and YbB_{12} . Second, we found that the charge-neutral fermions are strongly influenced by the underlying spin degrees of freedom. The coupling between neutral fermions and spin degrees of freedom has not been considered in existing theories.

-- --

Also there appears to be a typo when they write:

"We find that κ/T for $\forall \mu_0 H=0, 1, \text{ and } 2T$ is well fitted as $\kappa/T = \kappa_0/T + c_1 T^p$."

this should probably be " $\kappa/T = c_0 + c_1 T^p$ ", otherwise it will imply a finite heat conductivity at zero temperature which is not thermodynamically sensible (and not a finite κ/T).

Response:

We revised the expression as suggested by Reviewer #2.

With the above changes, we trust you will find our manuscript is much improved and is now suitable for publication in *Nature Communications*.

Sincerely yours,

Authors

REVIEWER COMMENTS

Reviewer #1 (Remarks to the Author):

The authors have made efforts to address some of the technical questions and data analysis issues that were highlighted in the first round of referee reports. I consider their responses as follows:

1. Contact resistance and thermal leakage.

The inclusion of the information about the measurements of the WF law on a stainless steel wire is a useful and reassuring test of thermal leakage problems with the measurement apparatus. The authors go on to say that it is difficult to measure the electrical conductance of the contacts at low temperatures due to the insulating nature of the sample.

A useful check of the relative size of the thermal conductance of the sample compared to the thermal conductance of the contacts is to plot $(T_+ - T_-)/(T_- - T_0)$ where T_+ and T_- are the hot and cold thermometers and T_0 is the temperature of the cold finger. This essentially measures the thermal gradient across the sample, compared to the thermal gradient across the current contact and can reveal if there are issues with thermal contacts at low temperatures. Are the authors able to provide access to this data for one of their in-field temperature sweeps where a downturn is observed?

2. Field dependence of the calibrations

The use of a Coulomb blockade thermometer to calibrate the RuO₂ thermometers in field is sensible and reassuring. It would be useful if an explanation of this procedure is included in the manuscript or the supplementary information.

3. Power law fitting and extrapolations.

A key issue with the current work involves the power law fits and extrapolations of the thermal conductivity. In order to extract a finite value of κ_0/T , the authors apply a floating power law fit to data taken in the AF-I phase, without a clear reason why. Do the authors have any suggestions as to why itinerant quasiparticles should follow a power law in this regime?

The authors then revert to a T^2 extrapolation in the AF-II region, dismissing a downturn in the data and using data only above the downturn temperature. This is justified by the proposed existence of a gap in the itinerant quasiparticle excitations, which seems highly speculative to me. It is not clear what causes the power law to change back to T^2 , and what might causes a gap to open.

Given that the low temperature thermal conductivity data plays a part in their argument for itinerant neutral quasiparticles, it is important that the extrapolations be robustly justified.

Reviewer #2 (Remarks to the Author):

The authors have accounted for the magnon contribution more carefully in this updated manuscript. They have also studied the sample dependence of specific heat and heat conductivities. With these improvements I find that the paper is in a satisfactory stage for publication.

Reviewer #1 (Remarks to the Author):

The authors have made efforts to address some of the technical questions and data analysis issues that were highlighted in the first round of referee reports. I consider their responses as follows:

Response:

We thank Reviewer #1 for careful reading of our manuscript and insightful comments, which helped us to further improve our paper.

1. Contact resistance and thermal leakage.

The inclusion of the information about the measurements of the WF law on a stainless steel wire is a useful and reassuring test of thermal leakage problems with the measurement apparatus. The authors go on to say that it is difficult to measure the electrical conductance of the contacts at low temperatures due to the insulating nature of the sample.

A useful check of the relative size of the thermal conductance of the sample compared to the thermal conductance of the contacts is to plot $(T_+ - T_-)/(T_- - T_0)$ where T_+ and T_- are the hot and cold thermometers and T_0 is the temperature of the cold finger. This essentially measures the thermal gradient across the sample, compared to the thermal gradient across the current contact and can reveal if there are issues with thermal contacts at low temperatures. Are the authors able to provide access to this data for one of their in-field temperature sweeps where a downturn is observed?

Response:

As suggested by the reviewer, we plot $(T_+ - T_-)/(T_- - T_0)$ vs. T at 3 T, where a downturn of κ/T is observed (Fig. R1). While κ/T significantly changes with T , $(T_+ - T_-)/(T_- - T_0)$ is nearly constant with T . Moreover, no discernible change of $(T_+ - T_-)/(T_- - T_0)$ is observed at the temperature where a downturn of κ/T is observed (the arrow in Fig. R1).

Fig. R1. T -dependence of $(T_+ - T_-)/(T_- - T_0)$, where T_+ , T_- , and T_0 are temperature of cold thermometer, hot thermometer, and cold finger, respectively, measured at 3 T. An arrow indicates the temperature at which the downturn of κ/T is observed. The temperature T is determined by the average of T_+ and T_- , $T = (T_+ + T_-)/2$.

2. Field dependence of the calibrations

The use of a Coulomb blockade thermometer to calibrate the RuO₂ thermometers in field is sensible and reassuring. It would be useful if an explanation of this procedure is included in the manuscript or the supplementary information.

Response:

We added the procedure of the thermometer calibration under magnetic fields in the method section of the revised manuscript.

3. Power law fitting and extrapolations.

A key issue with the current work involves the power law fits and extrapolations of the thermal conductivity. In order to extract a finite value of κ_0/T , the authors apply a floating power law fit to data taken in the AF-I phase, without a clear reason why. Do the authors have any suggestions as to why itinerant quasiparticles should follow a power law in this regime?

The authors then revert to a T^2 extrapolation in the AF-II region, dismissing a

downturn in the data and using data only above the downturn temperature. This is justified by the proposed existence of a gap in the itinerant quasiparticle excitations, which seems highly speculative to me. It is not clear what causes the power law to change back to T^2 , and what might causes a gap to open. Given that the low temperature thermal conductivity data plays a part in their argument for itinerant neutral quasiparticles, it is important that the extrapolations be robustly justified.

Response:

We have realized that the floating power law fit to the data in the AF-I phase may raise confusion. Therefore, in the revised manuscript, we plot κ/T as a function of T^2 in all the insets of Figs. 8(a)-8(h). The combination of κ/T vs. T (main panels) and κ/T vs. T^2 (insets) plots firmly confirms the presence of the finite residual T -linear term in the thermal conductivity at low fields because of the following reason. In the AF-I phase below 2.5 T, the T -dependence of κ/T shows a convex downward curvature for κ/T vs. T plot, while a convex upward curvature for κ/T vs. T^2 plot. Obviously, for both κ/T vs. T and κ/T vs. T^2 plots, the extrapolation of κ/T to $T = 0$ has a finite intercept at $\mu_0 H = 1$ and 2 T in the AF-I phase. This indicates that κ_{qp} contains a finite residual T -linear term $K_0 = \kappa_{qp}/T(T \rightarrow 0)$, providing evidence for the presence of neutral itinerant fermionic excitations. Thus, κ_{qp} is written as $\kappa_{qp}/T = K_0 + f(T)$. In the temperature range shown in Figs. 8(a)-8(h), thermal conductivity is given by the sum of phonon and quasiparticle contributions, $\kappa = \kappa_{qp} + \kappa_{ph}$, as the magnon thermal conductivity is negligibly small due to the magnon gap. We find that the T -dependent part of κ/T , i.e., $f(T) + \kappa_{ph}/T$, can be fitted by a power-law dependence on T , as depicted in the inset of Supplementary Figs. 5(a)-5(h). Then, to estimate K_0 accurately, we extrapolate κ/T to $T = 0$ by assuming the power-law T -dependence of $f(T) + \kappa_{ph}/T$. As for the gap formation in the AF-II phase, the downturn behavior of κ/T indicates that κ/T depends on T as $\kappa/T \sim T^q$ with $q < 1$. We stress that this T -dependence cannot be explained by phonons ($\kappa_{ph}/T \sim T^p$, $p \sim 2$). Therefore it is natural to consider that the downturn is attributed to the gap formation of the itinerant quasiparticle excitations. We explicitly described the above argument in the revised manuscript.

We also discuss the power law T -dependence of κ/T . As mentioned in the manuscript, the temperature and field dependencies are mainly determined by the quasiparticle contribution. However, the detailed T -dependence of κ_{qp} is difficult to determine due to the presence of a small but finite κ_{ph}/T . Although the origin of the neutral fermions in Kondo insulators remains elusive, our results demonstrate that the properties of neutral fermion excitations are largely affected by the underlying magnetic structure. The clarifications of the $f(T)$ -term and origin of the gap formation are important future issues to understand the coupling between the neutral fermions and spin degrees of freedom, which has never been considered in existing theories. We state these arguments in the revised manuscript.

With the above changes, we trust you will find our manuscript is much improved and is now suitable for publication in *Nature Communications*.

Sincerely yours,
Authors

REVIEWERS' COMMENTS

Reviewer #1 (Remarks to the Author):

The authors have in my opinion considerably improved the clarity of the presentation of the thermal conductivity measurements.

The information on thermal contact resistances is a good check for issues related to poor thermalisation and heat leaks at low temperatures, and the inclusion of the procedure for field calibration of the thermometer gives added confidence in the results.

I think that the reversion to a κ/T vs T^2 plot in the inset of figure 8 a)-h) is easier to interpret and understand. There could still be an alternative explanation for the data, but at least the authors arguments are now consistent and easy to follow. I expect other groups will (and should) attempt to reproduce these results in future experiments.

Overall, I am now satisfied that the paper is in a good enough position to be published in Nature Communications.